# Imaging and Deep Learning Based Approach to Leaf Wetness Detection in Strawberry

**DOI:** 10.3390/s22218558

**Published:** 2022-11-07

**Authors:** Arth M. Patel, Won Suk Lee, Natalia A. Peres

**Affiliations:** 1Department of Electrical & Computer Engineering, University of Florida, Gainesville, FL 32611, USA; 2Department of Agricultural and Biological Engineering, University of Florida, Rogers Hall, 1741 Museum Road, Gainesville, FL 32611, USA; 3Department of Plant Pathology, Gulf Coast Research and Education Center, University of Florida, Wimauma, FL 33598, USA

**Keywords:** artificial intelligence, color imaging, leaf wetness, Strawberry Advisory System, strawberry diseases

## Abstract

The Strawberry Advisory System (SAS) is a tool developed to help Florida strawberry growers determine the risk of common fungal diseases and the need for fungicide applications. Leaf wetness duration (LWD) is one of the important parameters in SAS disease risk modeling. By accurately measuring the LWD, disease risk can be better assessed, leading to less fungicide use and more economic benefits to the farmers. This research aimed to develop and test a more accurate leaf wetness detection system than traditional leaf wetness sensors. In this research, a leaf wetness detection system was developed and tested using color imaging of a reference surface and a convolutional neural network (CNN), which is one of the artificial-intelligence-based learning methods. The system was placed at two separate field locations during the 2021–2022 strawberry-growing season. The results from the developed system were compared against manual observation to determine the accuracy of the system. It was found that the AI- and imaging-based system had high accuracy in detecting wetness on a reference surface. The developed system can be used in SAS for determining accurate disease risks and fungicide recommendations for strawberry production and allows the expansion of the system to multiple locations.

## 1. Introduction

Fungal diseases, such as botrytis fruit rot and anthracnose, are major contributors to the yield loss in strawberry crop [1,2]. Every season, farmers spray fungicides regularly to prevent the proliferation of these diseases. These fungicides increase production costs and contribute to a decrease in profit. The proliferation of these fungal diseases is directly linked to the presence of free water on the plant canopy, called leaf wetness [3]. By measuring the duration for which the water is present, i.e., leaf wetness duration (LWD), along with the temperature, the risk factors of those diseases can be derived and farmers can be informed to spray fungicides exactly when they are needed. This avoids spraying fungicides when they are not needed, leading to savings in production costs and a reduction in the selection pressure for resistance.

The Strawberry Advisory System (SAS) is one such tool developed to help Florida strawberry growers with knowledge of disease risk factors in their location and fungicide application recommendations [4]. SAS uses data from traditional flat-plate leaf wetness sensors to measure the LWD. In places where conventional leaf wetness sensors are not installed, it uses weather data, such as temperature, humidity, and dew point. These meteorological data are used in various mathematical models to predict water presence and estimate the LWD [5]. There are a few limitations of these methods, which prevent their wider use. For example, leaf wetness sensors need to be painted and calibrated from time to time [6], and the mathematical models have different accuracies and are not highly reliable [3]. In addition to these two methods, there have been several other attempts made to detect leaf wetness, such as paper-based chip RFID sensors [7], cylindrical sensors [8], cloth-based electrical resistance sensors [9], and beta ray gauge sensors [9]. Each of these sensors has some limitations, making it less adaptable to the varying nature of field conditions. Thus, there is a need for a better tool to detect leaf wetness [10].

A feasibility study was carried out to detect leaf wetness using an imaging-based device [11]. This study concluded that a device could be made that takes images of a reference surface, and using a pre-trained convolutional neural network model, the images can be classified into wet/dry categories. Thus, the LWD can be measured without using traditional leaf wetness sensors. The objective of this study was to develop in-field imaging-based leaf wetness detection systems and evaluate their performance. Two such systems were developed and placed at two field locations for strawberry production. The results were evaluated using manual observations and SAS data (SAS, http://cloud.agroclimate.org/tools/sas/dashboard/disease, accessed on 30 May 2022).

## 2. Materials and Methods

A leaf wetness detection system was developed and placed at the Plant Science Research and Education Unit (PSREU) of the University of Florida (UF) in Citra, Florida, USA, from October 2021 to March 2022. A similar system was set up at the Gulf Coast Research and Education Center (GCREC) of the University of Florida, Wimauma, Florida, USA, during the strawberry-growing season from February to March 2022. Both systems were placed approximately 5 m away from the strawberry-growing field.

Figure 1 shows the system setup at UF PSREU, and Figure 2 shows the system setup at UF GCREC. Both systems contained a color camera, a painted reference surface, LED light for artificial illumination during the night, a single-board computer, solar panels, and a battery for power supply. Details of these hardware components are described later. Figure 3 shows a block diagram of the system. Though conceptually the same, there was only one difference between the two systems, the orientation of the artificial illumination. As shown in Figure 1, when the artificial illumination was from behind, extra components and space were needed, making the system larger. So, it was better to add artificial illumination on the front side to save some space. The orientation of the artificial illumination, whether front or back, did not make a difference in the image outcomes as long as the artificial illumination was aligned with the installation angle of the reference surface.

### 2.1. Reference Surface

The reference surface represents a leaf on the field. A 25.4 × 20.3 cm acrylic sheet painted with flat white paint (Rust-Oleum American Accents 2X Ultra Cover Flat Spray Paint, Rust-Oleum, Evanston, IL, USA) was used as a reference surface. The reference surface was placed 45 cm above ground level, facing north, at an angle of 45° from the ground, following recommendations for the placement of leaf wetness sensors [12].

Before setting up the system on the field, flat surfaces with various materials and colors were used to experiment with the best color image outcome in the presence of water on the surface. The materials tried were polypropylene sheets with matt and gloss finish of various colors, wooden and metal surfaces painted with flat white paint, and acrylic sheets painted with flat white paint. Of these, the acrylic sheets yielded the best detection results for water droplets.

Since water droplets slip easily on surfaces with glossy textures, this might cause a reduction in the overall LWD; hence, surfaces with glossy textures were eliminated. Colored surfaces with dark colors absorb more sunlight and can become warm quickly, which might also reduce the overall LWD, so they were also eliminated. Finally, a surface painted with flat white paint, which does not create a glossy surface, was used as a reference surface. The acrylic sheet was a better choice because, compared to wood, it had a uniform texture and better endurance in outdoor conditions. In addition, unlike metal, it does not become warm.

### 2.2. Color Imaging of the Reference Surface

An RGB camera (WYZE Cam v2, WYZE Labs, Seattle, WA, USA) with a resolution of 1920 × 1080 pixels was used to take images of the reference surface. The images were taken every 15 min. The camera was placed inside a weatherproof enclosure 50 cm above ground, facing the reference surface and in a north-south direction. The distance between the reference surface and the camera enclosure was 20 cm. Figure 4 shows the camera setup at UF PSREU.

The camera was connected to a single-board computer (Raspberry Pi 4, Raspberry Pi Foundation, Cambridge, UK) using a USB cable. The captured images were sent to Google Drive (Google LLC, Mountain View, CA, USA) for data analysis and storage. A wireless cellular modem (Verizon Jetpack MiFi 8800L, Verizon Communications Inc., New York City, NY, USA) was used to transfer field images to Google Drive. Figure 5 shows example images taken by the camera under rainy conditions during the day and at nighttime.

### 2.3. Artificial Illumination

To take images during the night, a 12V LED light (AJ-Ultra thin Eagle Eye, AUTOMONARCH, Shenzhen, China) was used for artificial illumination.

The light was placed at a 45° angle in alignment with the reference surface. The placement of the light was this way to create a shadow of the water droplets present on the reference surface. Figure 6 shows images taken during the night using this arrangement of artificial illumination.

### 2.4. Image Processing

The original images of the reference surface had a resolution of 1920 × 1080 pixels and contained the entire reference surface, which had a size of 25.4 × 20.3 cm. As shown in the original image in Figure 7a, the camera lens had barrel distortion present. The barrel distortion was corrected programmatically. Figure 7a shows the original image with barrel distortion, and Figure 7b shows the corrected image.

After correcting distortion, the images were cropped to 300 × 200 pixels from the center of the reference surface, representing a 7.6 × 5 cm surface. This size was chosen because the currently available wetness sensors in the market have approximately the same size. The cropped images were used to train and test the convolutional neural network model.

### 2.5. Training and Test Datasets

#### 2.5.1. Training Dataset

The color image training datasets contained 25,000 unique color images of the reference surface. These images were taken at UF PSREU, Citra. The images were selected randomly from the images taken from September through November 2021. The images were from the day and nighttime. Each image was manually observed and assigned either a “wet” or a “dry” label. Figure 8 shows example images. There were 13,663 images under the “wet” class and 11,337 images under the “dry” class. The dataset was biased toward the “wet” class. To avoid bias in the final model, class weights were adjusted while training the convolutional neural network. The “dry” class was assigned 1.20 times the weightage of the “wet” class images. The dataset was divided into training and validation sets in an 8:2 ratio. The training set contained 20,000 images; the validation set contained 5000 images.

#### 2.5.2. Test Dataset

There were two test datasets. Test set 1 contained images taken at UF PSREU, Citra, from December 2021 to March 2022. This dataset contained 19,000 unique color images of the reference surface. Test set 2 contained images taken at UF GCREC, Wimauma, from February to March 2022. This dataset contained 1100 images. All the images were labeled manually using visual observation and assigned a “wet” or a “dry” label based on the presence of water on the surface. The test sets were used to check the accuracy of the deep learning model.

### 2.6. Deep Learning Algorithm

A convolutional neural network (CNN) was used to automatically classify images of the reference surface into two classes. A CNN usually performs better in image classification problems when a large amount of data are available. In addition, previous studies [11,13] have tried image processing and other approaches using color and thermal images, but those techniques had limitations and did not yield promising results. Hence, a CNN was a preferred choice for our task.

In this study, a sequential CNN model was used, which had one input layer, hidden layers, and one output layer. In this case, the input layer was a vector with a size of (200, 300, 3), which is the size of the input image. The hidden layers had a series of convolution and max pool layers, followed by a series of activation layers. The output layer was a vector of size (10), using the ReLU activation function, and the image class was determined from the output layer. Figure 9 shows the details of the CNN layers used to classify the reference surface images into two classes. The conceptual understanding of the CNN architecture is provided in [14].

The model was compiled using an optimizer. In this case, the binary cross-entropy function was used. The class weights were adjusted before training the CNN to avoid bias in the trained model. Additionally, dropout layers, early stopping, and data hold-out methods were used to avoid the overfitting problem. Dropout is a regularization method that randomly drops several output layers, hence reducing the complexity of the neural network and avoiding overfitting. Early stopping was used to stop further training of the neural network once the maximum accuracy was reached. In addition, 20% of the data in the training dataset were used for validation, which was to ensure that the model did not overfit while training the neural network. The model was trained using a training set. The validation set was used to check the accuracy during the training. The training and testing of the CNN were conducted using Google Colab Notebook (Google Colaboratory, Google LLC, Mountain View, CA, USA). The model was trained for 50 epochs. Figure 10 shows the training and validation accuracy trend as the model was being trained.

### 2.7. Evaluation Methods

Several evaluation matrices were used to evaluate the performance of the trained neural network. For the trained model, accuracy, precision, and recall were determined.
(1)$$\mathrm{Accuracy}=\frac{\mathrm{TP}+\mathrm{TN}}{\mathrm{TP}+\mathrm{TN}+\mathrm{FP}+\mathrm{FN}}$$
(2)$$\mathrm{Precision}=\frac{\mathrm{TP}}{\mathrm{TP}+\mathrm{FP}}$$
(3)$$\mathrm{Recall}\;=\frac{\mathrm{TP}}{\mathrm{TP}+\mathrm{FN}}$$

Here, TP denotes true positives (i.e., “wet” class image predicted as “wet”), TN denotes true negatives (i.e., “dry” class image predicted as “dry”), FP denotes false positives (i.e., “dry” class image predicted as “wet”), and FN denotes false negatives (i.e., “wet” class image predicted as “dry”).

## 3. Results

The trained CNN model was used to predict labels for the images in test set 1 and test set 2. The predicted labels were compared with the manually assigned labels. Table 1 shows the accuracy, precision, and recall for both test sets when the results were compared with the visual observation of the images of the reference surface. Table 2 and Table 3 show the confusion matrix for test set 1 and test set 2, respectively.

To get a better perspective on the model’s accuracy, the predicted labels were also compared with SAS data. Test set 1 results were compared with SAS’s UF PSREU, Citra, data, and test set 2 results were compared with SAS’s UF GCREC, Wimauma, data. SAS’s PSREU, Citra, station uses weather data and mathematical models, and the GCREC, Wimauma, station uses two flat-plate electronic leaf wetness sensors (L-237, Campbell Scientific, Inc., Logan, UT, USA) to detect wetness.

SAS collects data every 15 min, and the new proposed system in this study also collected data every 15 min. However, there were occasions when SAS data were unavailable. It was due to the SAS wetness detection system not being active during those periods. So, there were 5458 instances where the data captured by the new system at PSREU were matched with the data captured by the existing SAS. For GCREC, there were 1051 such instances. As the data capture timings were the same, these results could be compared. Table 4 shows the accuracy, precision, and recall for both test sets. Table 5 and Table 6 show a confusion matrix for test set 1 and test set 2, respectively.

## 4. Discussion

It is clear from the results in Section 3 that the deep learning method yielded high accuracy when the model’s results were compared with the manual observation-labels. The results in Table 1 are high, but there is still room for improvement. There are a few factors that affect the gap in the model’s accuracy when predictions are compared with visual observations.

During the dew onset period, the water droplets were tiny (<0.01 mm) and were spread across the surface as a thin layer of water. This made it difficult to visually observe those water droplets and categorize and label these images correctly, as shown in the example image in Figure 11. Another factor that affected the mislabeling was when only one or two small droplets (~1–3 mm) were present on the reference surface. This situation usually occurred during the dew offset period. It was difficult to categorize these images into “wet” or “dry” categories, as shown in Figure 12. These errors in the labeling of the images were carried forward into the trained model and eventually contributed to the gap in the accuracy of the test datasets.

These inaccuracies do not contribute to a significant drop in the overall LWD. They contribute to less than ±1.5 h in the overall LWD for a given day. The current SAS continues accumulating the LWD until there is a gap of at least 4 h after the end of a wetness period [15]. The inaccuracy in the new system can be tolerated, given the varying nature of infield conditions, since currently used sensors are also problematic during the onset and offset of wetness periods and thresholds need to be determined during calibration.

The results of the CNN model’s prediction in Table 4 are significantly lower compared to the results in Table 1. SAS’s Citra weather station uses various meteorological data, such as temperature, humidity, and solar radiation, in mathematical models, such as the Dew Point Depression (DPD), the Classification and Regression Tree (CART), the Number of Hours with Relative Humidity equal or greater than 90% (NHRH ≥ 90%), and the Penman-Monteith (PM) model, to predict the wetness period. These models have limitations [2], so their results can differ from the results produced by the visual observation-based model. In addition, SAS’s Citra weather station is located approximately 3 km away from where the wetness detection system used in this study was set up. Thus, this distance could also have contributed to the differences in the results.

SAS’s GCREC weather station uses calibrated leaf wetness sensors to detect leaf wetness, and this station was located approximately 5 m away from the system used in this study. A comparison between the leaf wetness sensor method and the imaging-based method was made in [11], which explains the differences in the results.

There are several things to consider regarding the reference surface. The size of the reference surface (25.4 × 20.3 cm) was chosen arbitrarily, but the images used in the training and testing of the model had a size of 7.6 × 5 cm. This is approximately the same size as the currently available electronic leaf wetness sensors in the market. The variance in the size of the reference surface can have an effect on the LWD period. If the size of the reference surface is too large, then it can collect more water, eventually increasing the LWD. With a larger size reference surface, it can also collect dust, bird droppings, and some other materials that might be detected as water droplets, creating false positives. A larger reference surface size can also interfere with other farm equipment and damage it. If the size of the reference surface is too small, then it might miss periods of actual wetness and might create false negatives. Thus, we believe that the proper size of the reference surface should be approximately 5–10 cm × 5–10 cm.

In this study, the reference surface was painted with flat white paint. As the paint withers over time, there was a need to paint the reference surface periodically, e.g., every 8 weeks. In the future, a white surface with a non-glossy and non-reflective texture can be used to avoid the need of painting the reference surface. An automatic reference surface-cleaning mechanism, such as a wiper, can also be attached to avoid any need to clean reference surfaces manually, avoiding any need for maintenance.

In the future, the above-mentioned improvements can be made to the existing system. Since this system is highly accurate, it can be placed at multiple locations and data from these systems can be directly used in SAS in real time to help strawberry growers in Florida with an accurate estimation of LWD periods, disease risk factors, and fungicide recommendations.

## 5. Conclusions

This study developed a new leaf wetness detection system and tested an approach for leaf wetness detection using color imaging of a reference surface and deep learning. This approach yielded high accuracy when compared to visual observation labels of the corresponding images. For the GCREC system, the results were compared with wetness sensor data, which also had high accuracy. This model had low accuracy when results were compared with a weather-data-based approach to detect leaf wetness. Overall, this system has good potential and can be used in place of an electronic leaf wetness sensor. With few adjustments to the current system, it can be made maintenance free, and data can be used in SAS for strawberry disease risk factor calculations and fungicide spray schedule recommendations.

## Figures and Tables

**Figure 1 sensors-22-08558-f001:**
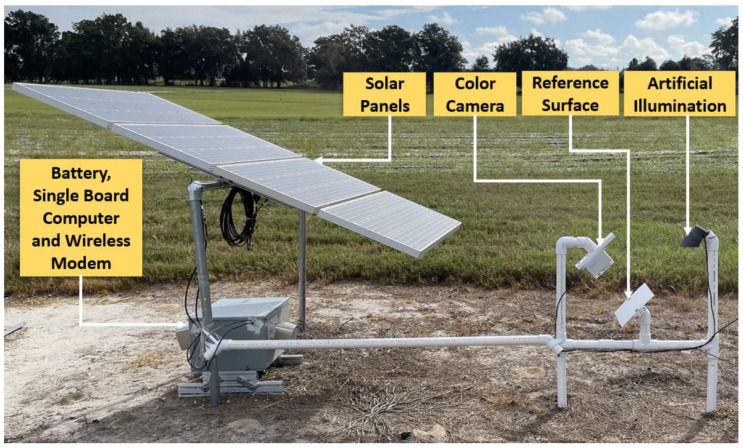
System to monitor a reference surface using an RGB camera at UF PSREU, Citra.

**Figure 2 sensors-22-08558-f002:**
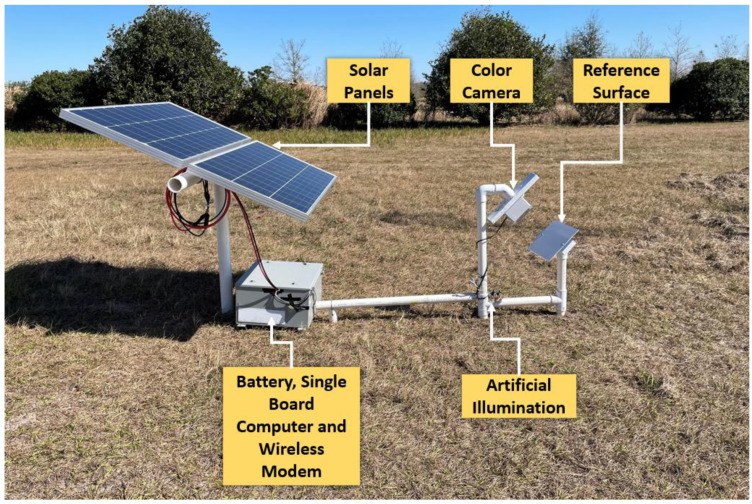
System to monitor a reference surface using an RGB camera at UF GCREC, Wimauma.

**Figure 3 sensors-22-08558-f003:**
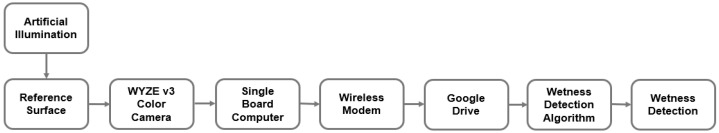
Block diagram of the system to monitor a reference surface and to detect wetness from images of the reference surface.

**Figure 4 sensors-22-08558-f004:**
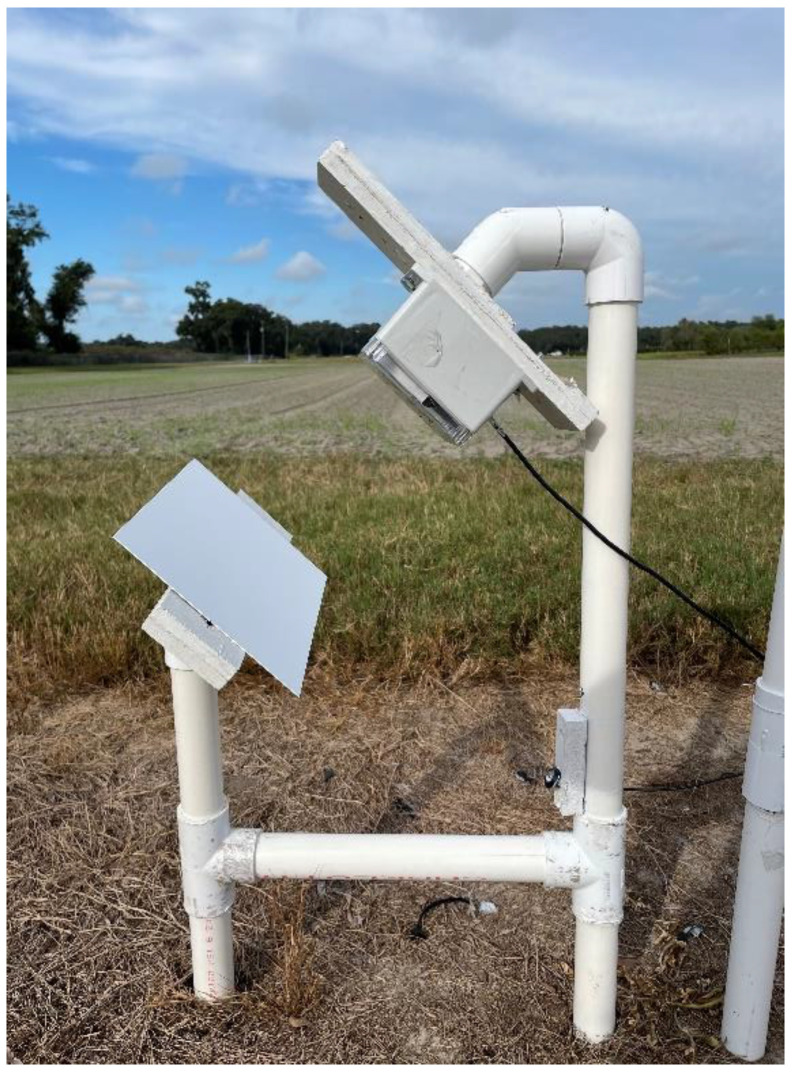
Reference surface and camera enclosure with the RGB camera at UF PSREU, Citra.

**Figure 5 sensors-22-08558-f005:**
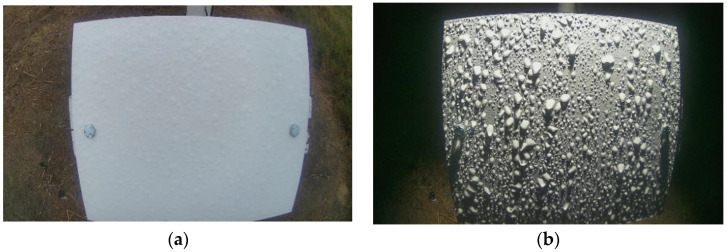
Example images of the reference surface: (**a**) color image acquired during normal daylight conditions and (**b**) color image acquired during the nighttime with the help of artificial illumination.

**Figure 6 sensors-22-08558-f006:**
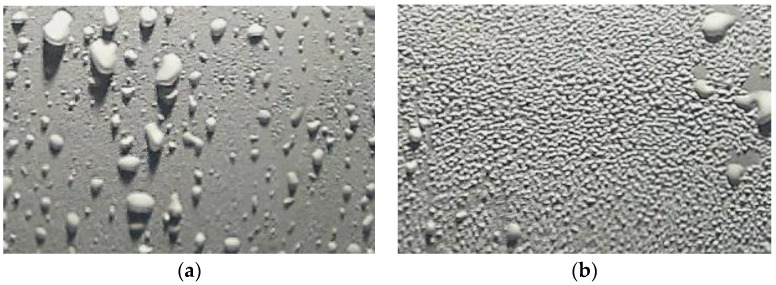
Example nighttime images of the reference surface using artificial illumination: (**a**) with large water droplets formed due to rain and (**b**) with tiny water droplets formed due to dew.

**Figure 7 sensors-22-08558-f007:**
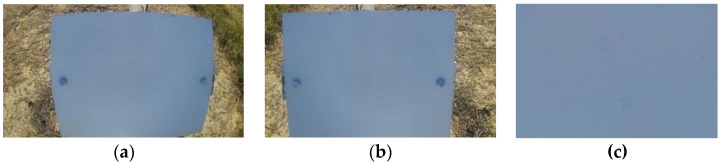
Example images of the reference surface: (**a**) the original image of the reference surface, (**b**) image with corrected barrel distortion, and (**c**) cropped image, which represents a 7.6 × 5 cm surface.

**Figure 8 sensors-22-08558-f008:**
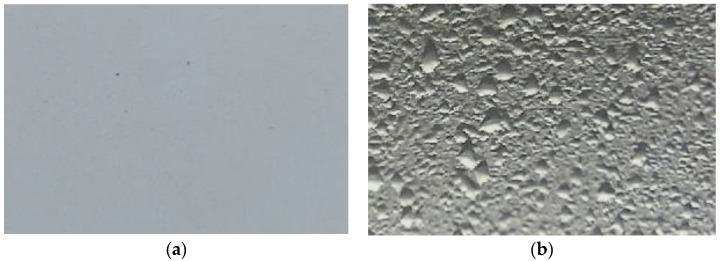
Examples of images used in the training and test datasets: (**a**) “dry” class image and (**b**) “wet” class image.

**Figure 9 sensors-22-08558-f009:**
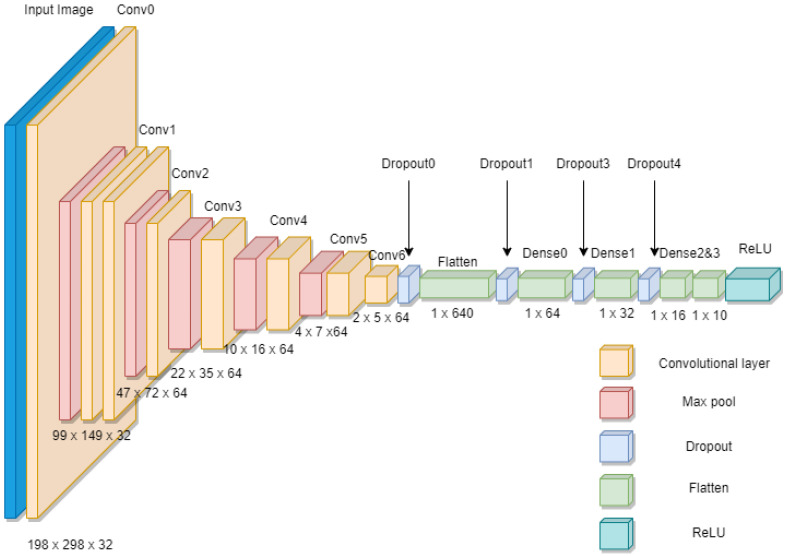
Details of the convolutional neural network layers used for the classification of the images of the reference surface into two classes.

**Figure 10 sensors-22-08558-f010:**
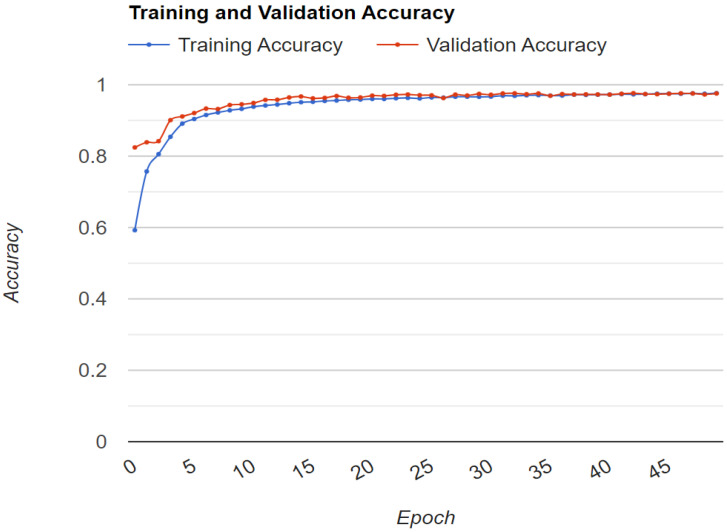
Training and validation accuracy trend.

**Figure 11 sensors-22-08558-f011:**
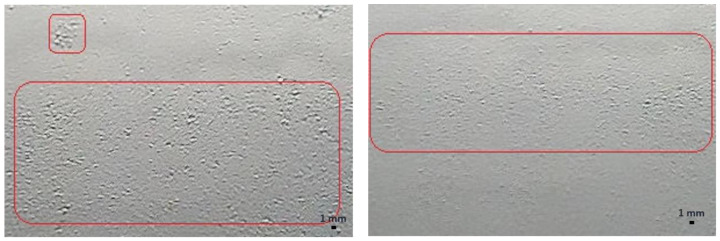
Example images of the reference surface with tiny water droplets during the dew onset period, shown in a red square.

**Figure 12 sensors-22-08558-f012:**
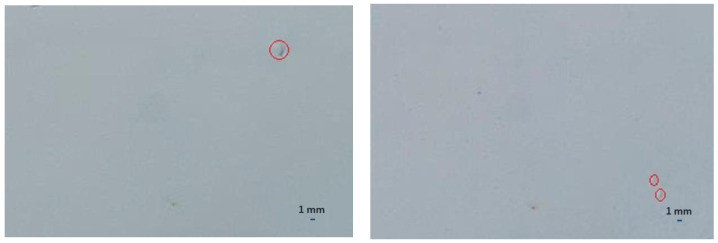
Example images of the reference surface with one water droplet, shown in a red circle.

**Table 1 sensors-22-08558-t001:** Results of the CNN model’s prediction when compared with visual observation of the reference surface images.

	*Test Set 1*	*Test Set 2*
*Accuracy*	0.962	0.954
*Precision*	0.946	0.932
*Recall*	0.962	0.944

**Table 2 sensors-22-08558-t002:** Confusion matrix for test set 1 when results were compared with manually assigned labels.

		**True Labels**			
		**“Wet”**	**“Dry”**		
**Predicted Labels**	**“Wet”**	0.962	0.038		1
					0.8
					0.6
	**“Dry”**	0.038	0.962		0.4
					0.2
					0

**Table 3 sensors-22-08558-t003:** Confusion matrix for test set 2 when results were compared with manually assigned labels.

		**True Labels**			
		**“Wet”**	**“Dry”**		
**Predicted Labels**	**“Wet”**	0.944	0.041		1
					0.8
					0.6
	**“Dry”**	0.056	0.959		0.4
					0.2
					0

**Table 4 sensors-22-08558-t004:** Results of the CNN model’s prediction when compared with SAS data.

	*Test Set 1*	*Test Set 2*
*Accuracy*	0.793	0.922
*Precision*	0.838	0.876
*Recall*	0.705	0.913

**Table 5 sensors-22-08558-t005:** Confusion matrix for test set 1 when results were compared with SAS PSREU data.

		**True Labels**			
		**“Wet”**	**“Dry”**		
**Predicted Labels**	**“Wet”**	0.706	0.128		1
					0.8
					0.6
	**“Dry”**	0.294	0.872		0.4
					0.2
					0

**Table 6 sensors-22-08558-t006:** Confusion matrix for test set 2 when results were compared with SAS GCREC data.

		**True Labels**			
		**“Wet”**	**“Dry”**		
**Predicted Labels**	**“Wet”**	0.913	0.073		1
					0.8
					0.6
	**“Dry”**	0.087	0.927		0.4
					0.2
					0

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
