# Peer review of "Imaging and Deep Learning Based Approach to Leaf Wetness Detection in Strawberry"

_sensors, 2022, doi:10.3390/s22218558_

Round 1

Reviewer 1 Report

Overall, it is an interesting task. The task can be further improved by

The authors do not demonstrate following the writing standards, and formatting errors can be found in several parts of the work. Because they are a lot, it would be a long list for me to list them one by one, so I suggest that the authors read the work very carefully.

The Abstract could tell what kind of deep learning is used.

The manuscript paper is not well prepared either. In the related work section, a lot of statements are not related to the topic of this paper. The figures are not well prepared/described and the presented results didn’t support the conclusion claimed by the authors.

Author Response

Thank you for your time and effort in providing valuable comments to improve the quality of the manuscript! We tried our best to address all your comments below. The revised portion of the text in the manuscript is shown in red

Reviewer #1:

Comment [1]: The authors do not demonstrate following the writing standards, and formatting errors can be found in several parts of the work. Because they are a lot, it would be a long list for me to list them one by one, so I suggest that the authors read the work very carefully.

Response: We tried to remove all the formatting errors that we can find. For example, all the images are now aligned, some of the tables have been replaced, and a few other minor formatting errors were fixed. Kindly suggest in case we missed considering any such required changes.

Comment [2]: The Abstract could tell what kind of deep learning is used.

Response: This has been corrected in lines 18-19.

Comment [3]: The manuscript paper is not well prepared either. In the related work section, a lot of statements are not related to the topic of this paper. The figures are not well prepared/described, and the presented results didn’t support the conclusion claimed by the authors.

Response: In the introduction section, we tried to give some general background of what different attempts were made to detect leaf wetness. Since leaf wetness is not a well-defined concept, there is no standard way to measure it. The proposed method is a novel approach, and we did not find any previous such approach to the problem of leaf wetness detection.

All the figures are corrected according to the manuscript guidelines.

We concluded that leaf wetness could be detected using an imaging and deep learning based system such as the one proposed in this paper because it has high accuracy. The results in Table 1 show 96.2% accuracy for a dataset with 19,000 images taken over several months, while comparing the model’s predicted results with the manually assigned labels. These results show that this system is robust and has high accuracy. Kindly point out the part of the conclusion which is not supported by the results.

Reviewer 2 Report

- Manuscript discusses the detect wetness from images of the reference surface, which is appreciated.

- Authors are suggested to add one flow chart of proposed work with existing works in Introduction Section.

- In Sub-section 2.5, color image training datasets have been trained with 25K to reference surfaces (under "wet" and "dry" classes). But, it is recommended to add more accuracy to such experimentation, if found.

- In Sub-section 2.6, CNN can be experimented with different classifier model, Pls comment on these adaptability for proposed exprimentation.

- Cite below article to improve the readability of your paper:                   Prognostic Kalman Filter Based Bayesian Learning Model for Data Accuracy Prediction", S. Karthik, Robin Singh Bhadoria, Jeong Gon Lee, Arun Kumar Sivaraman, Sovan Samanta, A. Balasundaram, Brijesh Kumar Chaurasia, S. Ashokkumar, Journal Computers, Materials & Continua (CMC), Vol. 72, Issue 1, pp. 243-259, 2022.

Author Response

Thank you for your time and effort in providing valuable comments to improve the quality of the manuscript! We tried our best to address all your comments below. The revised portion of the text in the manuscript is shown in red

Reviewer #2:

Comment [1]: Manuscript discusses the detect wetness from images of the reference surface, which is appreciated.

Response: Thank you for the consideration of the manuscript.

Comment [2]: Authors are suggested to add one flow chart of proposed work with existing works in Introduction Section.

Response: Thank you for your suggestion. The proposed method builds on previous work (Patel et al., 2021) which used a similar approach to detect leaf wetness but using color and thermal imaging. In that case, instead of a reference surface, leaf wetness sensor images were taken. So, the block diagram shown in Figure 3 in the current manuscript is similar to the one shown in Figure 2 by Patel et al. (2021). The suggested flow chart will create an unnecessary repetition of information that is already covered in Figure 3. Kindly suggest if recommended changes still need to be made.

Reference:

[1] A. Patel, W.S. Lee, N. Peres, C. Fraisse. “Strawberry plant wetness detection using computer vision and deep learning”. Smart Agricultural Technology, (2021), vol 1, ISSN 2772-3755. https://doi.org/10.1016/j.atech.2021.100013

Comment [3]: In Sub-section 2.5, color image training datasets have been trained with 25K to reference surfaces (under "wet" and "dry" classes). But it is recommended to add more accuracy to such experimentation, if found.

Response: Thank you for your suggestion. We will consider this in our future study. We can add several other classes instead of just two classes.  For example, dew onset period, dew offset period, etc., or based on the amount of water present 10% wet surface, 20% wet surface, etc.

Comment [4]: In Sub-section 2.6, CNN can be experimented with different classifier model, Pls comment on this adaptability for proposed experimentation.

Response: We tried using other classifiers such as using image processing, decision tree, support vector machine (SVM), etc., but they did not perform well. Although the problem is simple two-class image classification, the images have high variability. For other classification methods, overfitting was a major issue, and it was hard to improve model accuracy despite the large dataset. We did not include those results because our focus was on wetness detection and the model which has potential.

Comment [5]: Cite below article to improve the readability of your paper:                   Prognostic Kalman Filter Based Bayesian Learning Model for Data Accuracy Prediction", S. Karthik, Robin Singh Bhadoria, Jeong Gon Lee, Arun Kumar Sivaraman, Sovan Samanta, A. Balasundaram, Brijesh Kumar Chaurasia, S. Ashokkumar, Journal Computers, Materials & Continua (CMC), Vol. 72, Issue 1, pp. 243-259, 2022.

Response: Thank you for your suggestion. The suggested work does not seem to be related to the method we implemented as we did not use Bayesian learning and Kalman fileting in the proposed work. Kindly suggest in case we missed referring to this article and used its content.

Reviewer 3 Report

The authors made a test experiment that includes a dataset collection for the wet/dry classification problem for applications to determine the risk for common fungal diseases and the need for fungicide applications. The authors used colored images to build a decision system for classification that utilizes CNN models. The overall presentation and the idea of the article sound well. However, some comments could be addressed to improve the quality of the paper.

1) More detailed literature overview should be added that should be showing the differences between the state of the art and this paper's idea.

2) Figures need to be smaller to make the article looks prettier. Especially images (1, 2, and 3). Figures (4 and 5) could be aligned in one row also for figure 7.

3) Table 1 could be replaced with a visualized structure like the figure in this image (https://miro.medium.com/max/720/1*BncGa1UaQiYbYkSjLc39BA.png). Please put all the images on the top of the page.

4) Table 2 could be replaced by visual confusion matrix like this (https://i.stack.imgur.com/ljypq.png). Same for tables 3, 4,5,6, and 7. The tabular form of the confusion matrix is not clear and not easy to read.

5) The authors are encouraged to show the results of using other than the CNN models. Maybe, showing results for SVM, Xboost, Decision tree classifier, and Random forest classifier (as conventional classical classifiers). Do we really need a CNN model to perform inference for this type of 2 classes problem?

6) Some simple image segmentation techniques could be used to predict the class type, like watershed segmentation for counting water droplets. I suggest if possible consider testing it.

7) could you please elaborate more on the choice of the CNN structure and how you managed to ensure no overfitting or bias due to the imbalanced dataset? You mentioned giving more weight to one of the classes. However, there are many techniques to ensure no overfitting

8) Some figures showing the Epochs, validation, training accuracy, recall, and F_1 scores could be added to visually examining the robustness of the CNN-trained model.

I would like to thank the authors for their effort in the data acquisition and the overall effort in employing AI techniques for smart agriculture which will help in the future adoption of the automated farming 5.0 paradigm shift.

I suggest addressing some of the comments before accepting the article for publication.

Best regards

Author Response

Thank you for your time and effort in providing valuable comments to improve the quality of the manuscript! We tried our best to address all your comments below. The revised portion of the text in the manuscript is shown in red.

Reviewer #3:

Comment [1]: The authors made a test experiment that includes a dataset collection for the wet/dry classification problem for applications to determine the risk for common fungal diseases and the need for fungicide applications. The authors used colored images to build a decision system for classification that utilizes CNN models. The overall presentation and the idea of the article sound well. However, some comments could be addressed to improve the quality of the paper.

Response: Thank you for your consideration of the manuscript. We tried to address all comments below.

Comment [2]: More detailed literature overview should be added that should be showing the differences between the state of the art and this paper's idea.

Response: Thank you for your comment! Since leaf wetness is not a very well-defined concept, there is no standard way to measure it. The proposed method is a novel approach, and we did not find any previous approach to the problem of leaf wetness detection. In the Introduction section, we tried to give the background of all previous attempts that were made to solve this problem. We did not find any method which claims to be standard practice against which we can compare our results. Currently, electronic leaf wetness sensors and weather model-based approaches are popular, hence we compared our model’s results with those approaches. Kindly suggest if we have missed considering any particular approaches in this paper.

Comment [3]: Figures need to be smaller to make the article looks prettier. Especially images (1, 2, and 3). Figures (4 and 5) could be aligned in one row also for figure 7.

Response: Thank you for your comment! Suggested changes have been made on pages 3, 5, 6, and 10.

Comment [4]: Table 1 could be replaced with a visualized structure like the figure in this image (https://miro.medium.com/max/720/1*BncGa1UaQiYbYkSjLc39BA.png). Please put all the images on the top of the page.

Response: Thank you for your comment! Suggested changes have been made in Figure 9 on page 7. Also, all the images were aligned on the top of the page.

Comment [5]: Table 2 could be replaced by visual confusion matrix like this (https://i.stack.imgur.com/ljypq.png). Same for tables 3, 4,5,6, and 7. The tabular form of the confusion matrix is not clear and not easy to read.

Response: Thank you for your comment! Suggested changes have been made on pages 9 and 10, except Tables 1 and 4, which are not confusion matrices.

Comment [6]: The authors are encouraged to show the results of using other than the CNN models. Maybe, showing results for SVM, Xboost, Decision tree classifier, and Random Forest classifier (as conventional classical classifiers). Do we really need a CNN model to perform inference for this type of 2 classes problem?

Response: Thank you for your comment! We tried using other classifiers such as using image processing, k-means, support vector machine (SVM), etc., but they did not perform well. Although the problem is simple two-class image classification, the images have high variability. For other classification methods, overfitting was a major issue, and it was hard to improve model accuracy despite the large dataset. We did not include those results because our focus was on wetness detection and the model which has potential.

Comment [7]: Some simple image segmentation techniques could be used to predict the class type, like watershed segmentation for counting water droplets. I suggest, if possible, consider testing it.

Response: Thank you for your comment! The previous studies (Swarup et al. (2020) and Patel et al. (2021)) tried image processing approaches on color and thermal images, but those techniques had limitations and did not yield promising results. We can certainly explore more on this and will try to consider these approaches in the future.

This has also been added to the manuscript line 191-194 on page 7 as shown below.

“Also, previous studies [13] and [11] tried image processing and other approaches using color and thermal images, but those techniques had limitations and did not yield promising results. Hence, CNN was a preferred choice for the current task.”

Comment [8]: could you please elaborate more on the choice of the CNN structure and how you managed to ensure no overfitting or bias due to the imbalanced dataset? You mentioned giving more weight to one of the classes. However, there are many techniques to ensure no overfitting

Response: Thank you for your comment! We added all the steps taken to avoid overfitting in lines 205-211 as shown below.

“Additionally, dropout layers, early stopping, and data hold-out methods were used to avoid overfitting problems. Dropout is a regularization method that randomly drops several output layers, hence reducing the complexity of the neural network and avoiding overfitting. Early stopping was used to stop further training of the neural network once maximum accuracy was reached. Also, 20% of data in the training dataset was used for validation, which is to ensure that the model was not overfitting while training the neural network.”

Comment [9]: Some figures showing the Epochs, validation, training accuracy, recall, and F_1 score could be added to visually examining the robustness of the CNN-trained model.

Response: Thank you for your comment! We would appreciate it if you could please specifically point out which figures you are suggesting to add.

Comment [10]: I would like to thank the authors for their effort in the data acquisition and the overall effort in employing AI techniques for smart agriculture which will help in the future adoption of the automated farming 5.0 paradigm shift.

Response: Thank you for your comment!

References:

[1] A. Swarup, W.S. Lee, N. Peres, C. Fraisse. “Strawberry Plant Wetness Detection Using Color and Thermal Imaging”. J. Biosyst. Eng. (2020), 45, 409–421. https://doi.org/10.1007/s42853-020-00080-9

[2] A. Patel, W.S. Lee, N. Peres, C. Fraisse. “Strawberry plant wetness detection using computer vision and deep learning”. Smart Agricultural Technology, (2021), vol 1, ISSN 2772-3755. https://doi.org/10.1016/j.atech.2021.100013

Round 2

Reviewer 1 Report

Revision have done properly, you may consider this article for publication, Thanks  

Author Response

Thank you for your comments! We appreciate your time and help!

Reviewer 3 Report

Thank you for the revised version of the paper. Please see the following comments:

1- Please make the confusion mattresses smaller and in one row.

2- Please add training and validation curves for accuracy performance metrics. 

Best regards

Author Response

Comment [1]: Please make the confusion mattresses smaller and in one row.

Response: Suggested changes have been made in Tables 2, 3, 5, and 6 on pages 9 and 10.

Comment [2]: Please add training and validation curves for accuracy performance metrics. 

Response: Suggested change has been made. Figure 10 has been added on page 8, showing training and validation accuracy trends over 50 epochs as the model was being trained. Also, lines 219-220 are added regarding Figure 10.

“The model was trained for 50 epochs. Figure 10 shows the training and validation accuracy trend as the model was being trained.”